# UNC119 regulates T-cell receptor signalling in primary T cells and T acute lymphocytic leukaemia

Youhani Samarakoon[1,2], Tamas Yelland[1], Esther Garcia-Gonzalez[4], Amauri da Silva Justo Junior[1,2], Mahnoor Mahmood[1,2], Anand Manoharan[2], Shaun Patterson[2], Valentina Serafin[5], Payam A Gammage[1,2], Sandra Marmiroli[5], Christina Halsey[2], Shehab Ismail[3,*], Edward W Roberts[1,2,*]

**T-cell receptor recognition of cognate peptide–MHC leads to the formation of signalling domains and the immunological synapse. Because of the close membrane apposition, there is rapid exclusion of CD45, and therefore LCK activation. Much less is known about whether spatial regulation of the intracellular face dictates LCK activity and TCR signal transduction. Moreover, as LCK is a driver in T acute lymphocytic leukaemia, it is important to understand its regulation. Here, we demonstrate a direct role of the ciliary protein UNC119 in trafficking LCK to the immunological synapse. Inhibiting UNC119 reduces localisation of LCK without impairing LCK phosphorylation and reduces T-cell receptor signal transduction. Although important for initial LCK reorganisation, activated CD8$^+$ T cells retained their ability to kill target tumour cells when UNC119 was inhibited. UNC119 was also needed to sustain proliferation in patient-derived T-ALL cells. UNC119 may therefore represent a novel therapeutic target in T acute lymphocytic leukaemia, which alters the subcellular localisation of LCK in T acute lymphocytic leukaemia cells but preserves the function of existing cytotoxic lymphocytes.**

## Introduction

T cells are central to the adaptive immune system where TCR signalling is key for T-cell development and function (1). TCR signalling is initiated when antigenic peptides bound to the MHC are engaged by TCRs (2, 3). After TCR engagement, the activated lymphocyte-specific protein kinase (LCK) phosphorylates the immunoreceptor tyrosine–based activation motif consensus sites within the TCR (2, 4). Once phosphorylated, immunoreceptor tyrosine–based activation motifs recruit the zeta-chain–associated protein kinase 70 (ZAP70), which is further phosphorylated by LCK to become active (2, 5). Activated

ZAP70 then phosphorylates downstream proteins in the TCR signalling pathway, initiating T-cell activation (2, 4). Ligand-bound TCRs initiate the formation of submicron regions, known as microclusters, where cytosolic signalling proteins also aggregate (6). Prolonged TCR–peptide MHC (pMHC) interactions between a T cell and an APC cause reorganisation of TCR-pMHC complexes and adhesion molecules into a circularly symmetric structure termed the immunological synapse (IS) (7, 8). This reorganisation of the cell membrane halts cell movement and facilitates the execution of T-cell effector functions (7, 8).

LCK activity, like that of other SRC family kinases, is controlled by the phosphorylation of two critical tyrosine residues. Phosphorylation at tyrosine 394 within the activation loop enhances LCK activity, whereas phosphorylation at tyrosine 505 inhibits it (9, 10). In T cells, LCK exists in four distinct states: active (pY394), inactive (pY505), primed (unphosphorylated), and double-phosphorylated (pY394 + pY505) (9, 10). Nevertheless, it is reported that up to 40% of LCK within resting cells is constitutively active and its activity is not directly altered by TCR engagement. Instead, it is proposed that the spatial organisation of LCK in relation to its substrates, other kinases, and phosphatases is crucial to its proper function and role in TCR signalling (10).

Therefore, understanding how LCK is transported to the plasma membrane and whether this process is regulated is a highly important question. Within 10 min of synthesis, LCK is transported to the plasma membrane via vesicular trafficking, whereupon it associates with CD4 or CD8 (11). Interestingly, when cells were treated with brefeldin A, which disrupts vesicular transport, a portion of LCK still managed to reach the plasma membrane (11). This observation led the authors to propose that LCK might also use an alternative, direct pathway to the plasma membrane, bypassing the traditional vesicular trafficking route. Furthermore, a significant fraction of intracellular, cytosolic or attached to transport vesicles, LCK is found in T cells suggesting a cytosolic route of transport of LCK to the plasma membrane (12). This cytosolic fraction is found despite the lipid modification of LCK molecules. LCK undergoes co- and

[1]CRUK Scotland Institute, Glasgow, UK   [2]School of Cancer Sciences, University of Glasgow, Scotland, UK   [3]Department of Chemistry, KU Leuven, Heverlee, Belgium   [4]Central Laser Facility, Science & Technology Facility Council, The Research Complex at Harwell, Rutherford Appleton Laboratory, Harwell Campus, Oxfordshire, UK   [5]Department of Biomedical, Metabolic and Neural Sciences, Cellular Signalling Unit, University of Modena and Reggio Emilia, Modena, Italy

Correspondence: youhani.samarakoon@cruk.cam.ac.uk; shehab.ismail.mohamed@kuleuven.be; ed.roberts@glasgow.ac.uk
*Shehab Ismail and Edward W Roberts are cosenior authors

post-translational modifications including N-myristoylation and palmitoylation, which are essential for the localisation of LCK on plasma membranes and RAB11-positive endosomes (13, 14). Palmitoyl acyltransferase DHHC21 has specificity for LCK and is localised on the plasma membrane, and a palmitoylation–depalmitoylation cycle was suggested to be responsible for the subcellular localisation of LCK (15). UNC119 is a myristoyl binding protein that plays a significant role in maintaining myristoylated proteins in the primary cilium through cycles of solubilising myristoylated proteins and release at the primary cilia by the release factor ARL3GTP, which is specifically activated by ARL13B in the primary cilium (16, 17, 18). Interestingly, T cells lack primary cilium (19), but recently, it was reported, in Jurkat cells, that UNC119 plays an important role in shuttling LCK to the plasma membrane and specifically to the IS (20) with UNC119 being proposed to activate RAB11 to traffic LCK-containing endosomes to the IS (13). Indeed, the IS has been suggested to have many similarities to the primary cilium with it described as a "frustrated cilium" in these unciliated T cells, suggesting mechanisms in the primary cilium may be informative when studying the IS (7, 21). Furthermore, it was reported that UNC119 might play a role in activating LCK in CD4 cells (22). However, it is unknown whether these mechanisms control LCK localisation in primary T cells, and whether this has any impact on LCK activity in primary T cells.

Questions about the TCR signalling cascade are also key in T acute lymphocytic leukaemia (T-ALL). T-ALL occurs in both children and adults, and accounts for ~10–15% of paediatric ALL and 25% of adult ALL cases (23). Current therapies include high-dose glucocorticoids as part of intensive multi-agent chemotherapy, and haematopoietic stem cell transplantation, but a substantial proportion of patients (around 25% of paediatric and 40% of adult) are refractory to primary treatment or relapse post-therapy. Conventional cytotoxic therapy also leads to profound effects on normal immune function with significant infectious morbidity and mortality during treatment (24). To date, nelarabine remains the only FDA-approved chemotherapy agent that targets specifically relapsed T-ALL (25), whereas small molecule inhibitors of proteins like BCL2 are showing some promise (26, 27). The use of novel immune-mediated strategies of targeting ALL cells such as chimeric antigen receptor T-cell therapy is hampered by concerns over fratricide of normal T cells leading to profound immunodeficiency (28). Because of these high relapse rates and the need to preserve normal T-cell function, understanding of the mechanisms behind TCR-associated signalling will be key to the development of novel therapeutic strategies (23).

LCK activity is also subverted during T-ALL development where unregulated signalling, even in the absence of a TCR, leads to uncontrolled T-ALL cell proliferation (29). Critically, T-ALL cells do not always possess a TCR, so there is no segregation of CD45 from the TCR, nor is there a mechanical signal driving CD3 reorganisation (30, 31). In addition, there are no obvious reported mutations in LCK (32), CD45, nor CD3, which could account for changes in LCK signalling. Despite this, LCK is considered a good therapeutic target in around 40% of T-ALL cases (33). Because of the lack of understanding of how LCK activity is modulated in T-ALL, understanding potential mechanisms of LCK regulation may also have implications for T-ALL. Furthermore, constitutive oncogenic LCK signalling within T-ALL cells has previously been associated with resistance to

corticosteroid treatment in patients (34, 35), and thus, exploring ways of modulating LCK signalling remains an important effort.

Here, we demonstrate that despite their lack of cilia, the ciliary proteins UNC119, ARL13b, and ARL3 are expressed in all subsets of human T cells. Furthermore, they are expressed in naïve, effector, and memory CD8 T cells, suggesting they may have roles in all stages of development. Using a model of ovalbumin-specific CD8 T cells, OTI cells, we show that inhibition of UNC119 in naïve CD8 T cells using the inhibitor squarunkin A leads to a reduction in LCK polarisation and TCR signalling; however, in activated T cells, this is insufficient to inhibit cytotoxicity. Because this suggested UNC119 may represent a druggable regulator of LCK activation, we investigated whether the UNC119 pathway was expressed in human T-ALL cell lines, finding that all three proteins were expressed in three separate cell lines. We show that inhibition of UNC119 leads to a reduction in ZAP70 phosphorylation and T-ALL proliferation. This was recapitulated with a genetic approach where UNC119 was inducibly knocked down in T-ALL lines. Finally, we demonstrate that UNC119 inhibition reduces the growth of primary patient-derived T-ALL samples in vitro. Thus, inhibition of UNC119 may represent a potential novel target to inhibit T-ALL growth while maintaining cytotoxic function of existing CD8 T-cell responses.

# Results

## UNC119, ARL3, and ARL13B are broadly expressed in human T-cell subsets

Previous work has looked at UNC119 in Jurkat cells, which demonstrated a role in LCK localisation (13, 20). Knowing this and given the important role of LCK in TCR signal transduction, we wanted to investigate how broadly *UNC119*, and *ARL3* and *ARL13b* are expressed in human T-cell subsets. To address this, we analysed publically available scRNA-sequencing data of mature human T cells obtained from reference 36. These represented CD3+-enriched cells from the blood, bone marrow, lung, and lymph node, and the cells were distributed across 12 main clusters (Fig 1A). The expression of lineage markers *CD3e*, *CD4*, and *CD8* showed that clusters 0–8 consist of T cells with clusters 9–11 likely representing contaminating cell types (Fig 1B). As such, we analysed the expression of *ARL3*, *ARL13B*, and *UNC119* in clusters 0–8 (Fig 1C), which showed that both CD4+ and CD8+ T cells express these transcripts, indicating that these proteins may have a general role in T-cell function. We then analysed data from a separate study, which examined expression in naïve, activated, and memory antigen-specific murine P14 CD8 T cells (37) in a model of lymphocytic choriomeningitis virus (Fig 1D). In this analysis, CD8+ T cells from this model of infection form seven different clusters (Fig 1D), which segregate broadly based on whether the CD8+ T cells are naïve to the virus (d0), at peak (d9), or at memory (d129) post-infection with d0 and d129 overlapping considerably and d9 showing distinct phenotypes (Fig 1D). Analysing the expression of various lineage markers in the clusters showed that CD8+ T cells in cluster 6 expressed no examined markers, while clusters 0 and 4, enriched in the d9 sample, expressed high levels of *Klrg1* and *Gzmb* indicating

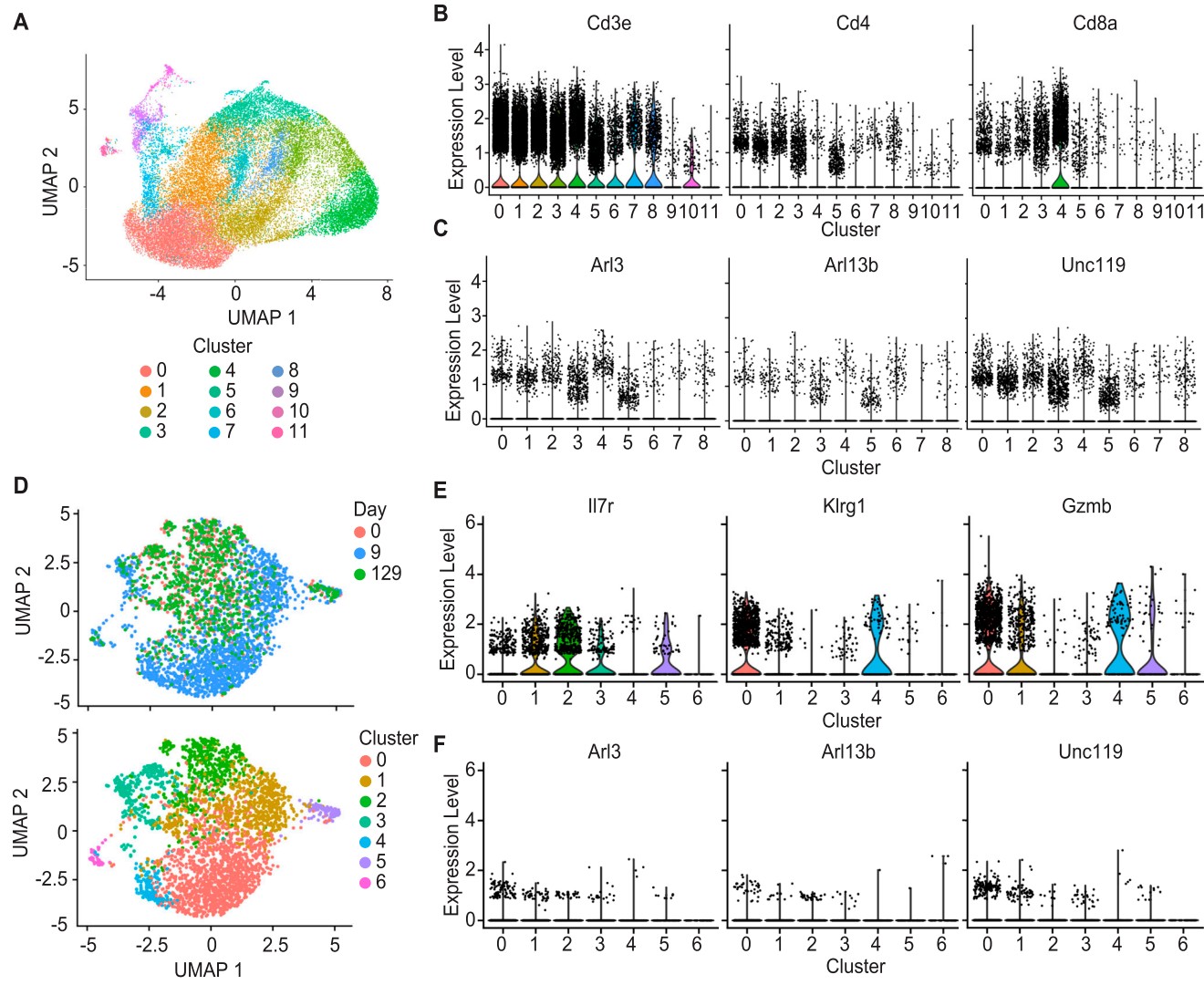

**Figure 1. UNC119, ARL3, and ARL13b are broadly expressed in multiple T-cell subsets.**
**(A)** UMAP of human scRNAseq of T cells from reference 36. **(B)** T-cell lineage markers as indicated on the right of the UMAP, indicating that clusters 1–8 are T cells. **(C)** Expression of ciliary *UNC119*, *ARL3*, and *ARL13b* in those T-cell clusters. **(D)** scRNAseq UMAP of P14 T cells from reference 36 after challenge with lymphocytic choriomeningitis virus (top), and clustering of cells at naïve (d0), peak (d9), and memory (d129) post-challenge (bottom). **(E)** Markers to identify cell phenotype based on *Il7r*, *Klrg1*, and *Gzmb* expression. **(F)** *Arl3*, *Arl13b*, and *Unc119* expression in clusters 1–6.

effector differentiation (Fig 1E), whereas the *Il7r* transcript was expressed more in the other clusters representing naïve and memory phases of infection (Fig 1E). What was striking was that CD8+ T cells in all clusters other than cluster 6 (Fig 1F) showed the expression of *Arl3*, *Arl13b*, and *Unc119*. This suggested to us that UNC119 has a functional role in all subsets of mature T cells and across different stages of differentiation. We therefore moved to study the expression and localisation of these at the protein level in primary CD8+ T cells.

### Unciliated primary CD8+ cytotoxic T cells express ciliary LCK binding and release factors

Having assessed the expression of *UNC119*, *ARL3*, and *ARL13b* transcripts in human CD8+ T cells, we asked whether this was also

true at the protein level and whether ciliary proteins have a role in unciliated primary CD8+ T cells. Primary naïve OTI T cells, expressing a TCR specific for the model antigen derived from chicken oval-bumin, were lysed and probed by Western blot analysis for Unc119, Arl3, and Arl13B, which were all observed in these naïve primary CD8+ T-cell lysates (Fig 2A).

To determine the subcellular localisation of these proteins during TCR signal transduction, OTI T cells were incubated with SIINFEKL-expressing target EG.7 lymphoma cells and imaged by confocal microscopy. Arl3 (Fig 2B) and Unc119 (Fig 2C) both showed a cytosolic distribution, with some overlap with the microtubule-organising centre (Fig 2B) in T cells. Near the microtubule-organising centre, there was also some indication of polarisation towards the membrane contacting the target cell (Fig 2B). UNC119 has been implicated previously in maintaining membrane-resident

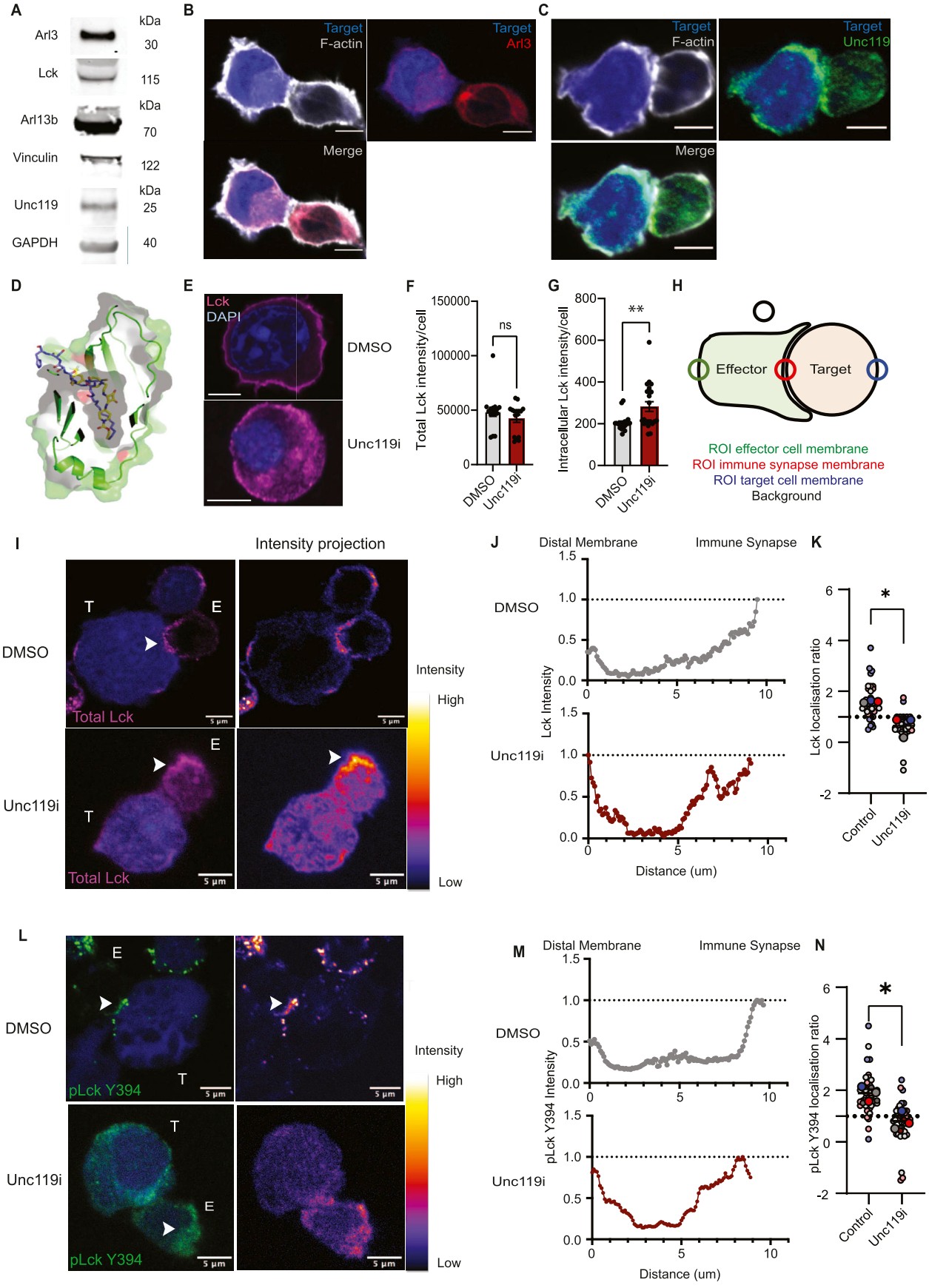

LCK in Jurkat cells (13), and given its expression within primary CD8+ T cells, it remained to determine whether this had functional significance. We therefore made use of a reported small molecule inhibitor of UNC119, squarunkin A (Unc119i) (38), to investigate the role of Unc119 in regulating LCK localisation, activation, and function within primary CD8+ T cells.

It has previously been reported that the small molecule squarunkin A, an UNC119i, is a direct competitor for UNC119 cargo proteins by binding to the UNC119 myristoyl binding site. We sought to confirm the mechanism of action of UNC119i by successfully determining the cocrystal structure of UNC119 with UNC119i to 2.21 Å resolution (Fig 2D). The structure confirmed the binding mode of squarunkin A (blue, Fig 2D) and its mode of action, which is indeed a direct competitor for the UNC119 myristoyl binding pocket (grey surface, Fig 2D). This is clearly observed when overlaying our new structure with that of UNC119 in complex with an N-myristoylated LCK peptide (PDB: 6H6A) (yellow, Figs 2D and S1A and B and Table S1) where significant steric clashes between the LCK peptide and UNC119i are seen. Thus, Unc119i provides a means to competitively inhibit LCK binding to Unc119 and could be used to determine whether the steady-state distribution of LCK in CD8+ T cells relies on Unc119 function. Primary naïve OTI T cells were stained for LCK in the presence of Unc119i or vehicle and imaged by confocal microscopy. Treatment with Unc119i led to a pronounced accumulation of intracellular LCK (Fig 2E). Quantifying LCK intensity within cells showed no change in total LCK levels (Fig 2F), confirming that this represented a redistribution of the LCK pool from the membrane to the cytosol (Fig 2G). It was next important to determine the role of Unc119 in TCR signal transduction, and so, we formed immune conjugates in vitro using OTI CD8+ T cells and SIINFEKL-expressing EG.7 lymphoma cells and quantified the polarisation of total LCK and pLCK Y394 to the IS as represented schematically (Fig 2H). Immune conjugates were stained for endogenous total LCK (red, Fig 2I) and the activating phosphate on pLCK, Y394 (green, Fig 2L), and visualised by confocal microscopy. Both total LCK and pLCK Y394 showed reduced polarisation to the IS with Unc119 inhibition, with vehicle-treated CD8+ T cells showing a peak of LCK and pLCK intensity at the IS (Fig 2J–M), whereas Unc119i-treated cells had broader distribution of LCK and pLCK throughout the cell and at the opposite membrane (Fig 2J–M). This change in distribution was subsequently quantified showing a significant reduction in polarisation upon inhibition of Unc119 (Fig 2K–N).

## Inhibition of UNC119 in primary CD8+ cytotoxic T cells reduces effector functions

To determine the functional consequences of reduced LCK polarisation to the IS in the absence of an impact on LCK phosphorylation, we first asked whether inhibition of Unc119 in CTLs affects conjugation to target cells in vitro. In in vitro–formed CTLs, early conjugate formation was unaffected by Unc119 inhibition, indicating that this is not required for TCR-mediated recognition of pMHC (Fig 3A). As such, we next quantified cytotoxic killing of lymphoma cells by Unc119-inhibited CTLs in vitro (Fig 3B). Surprisingly, there was no significant change in the ability of OTI CTLs to kill SIINFEKL-expressing target cells in the presence of Unc119i. Consequently, we sought to determine whether Unc119i would have an impact on LCK activity in naïve OTI cells during TCR signal transduction. To address this, OTI T cells were cocultured with splenocytes presenting SIINFEKL on H2-Kb in the presence of vehicle or Unc119i for 24 h, at which point samples were subjected to FACS analysis (Fig 3C). Flow cytometric analysis indicated no change in pLCK Y394 phosphorylation as previously suggested by the confocal microscopy analysis (Figs 3D and S2A–C), confirming that IS polarisation of LCK is not a prerequisite for its activating phosphorylation. There was, however, a trend towards a decrease in pLCK Y394 levels, suggesting that there may be some reduction that we were underpowered to detect. However, although LCK phosphorylation was not significantly impaired, phosphorylation of its downstream substrate, ZAP70, was significantly reduced in Unc119i-treated CTLs (Figs 3E and S2D–F). This indicated that in the absence of polarisation, LCK is unable to initiate downstream signalling. To confirm this, targets downstream of ZAP70 in the TCR signalling cascade were investigated. ZAP70 phosphorylates LAT to generate the LAT signalosome, which is a membrane-resident signalling hub coordinating downstream signalling, culminating in the activation of ERK. Phosphorylated ERK then translocates into the nucleus to

**Figure 2.  Primary CD8+ T cells express Unc119 at the protein level, which regulates LCK localisation at the steady state and during activation of the TCR.**
**(A)** Western blot analysis of the OTI T whole-cell lysate, showing the expression of Arl (top), Arl13b (middle), and Unc119 (bottom). Western blots were done separately, and each was blotted with an internal loading control: either LCK, vinculin, or GAPDH. **(B, C)** Confocal imaging of in vitro–formed immune conjugates between OTI T cells and EG.7 lymphoma cells showing the subcellular localisation of Arl3 (red) and F-actin (grey) to visualise the cell borders (B) and Unc119 (green) (C). Representative imaging of immune conjugates from two independent experiments, with OTI T cells isolated from n = 2 OTI mice. **(D)** Structure of an LCK peptide (PDB: 6H6A) overlayed with the UNC119i structure highlights binding to the same hydrophobic pocket (grey, indicated by text) of UNC119. N-myristoylated LCK peptide is shown in purple stick representation and UNC119i in yellow stick representation. Cross section of UNC119 is shown in green in cartoon and surface form. **(E)** Confocal imaging of endogenous LCK (magenta) in CD8+ T cells at the steady state with either vehicle (DMSO) or 5 μm Unc119i treatment. Representative images are from three independent experiments. **(F, G)** Quantifications of total intensity in vehicle- and Unc119i-treated CD8+ T cells (F), and quantification of membrane to cytosol redistribution of LCK, which was calculated by subtracting LCK intensity at the cell border from LCK intensity of the whole cell. Each dot on the graph represents an individual cell pooled from three individual experiments. **(H)** Graphical representation of how polarisation of LCK to the immune synapse was calculated. This was done using an ImageJ plugin called Synapse, referenced in the main text. **(I, J, K, L)** Confocal imaging of in vitro–formed immune conjugates was visualised for endogenous total LCK (red) and pLCK Y394 (green) with vehicle or 5 μm Unc119i treatment, shown on the left of both panels. Intensity projections on the right of both panels show intensity of LCK and pLCK at the IS and the rest of the T-cell membrane. Representative images are from three independent experiments of OTI T cells expanded from n = 3 OTI mice. **(J, M)** Quantification of LCK (J) and pLCK (M) through a section of a CD8+ T cell with vehicle or Unc119i treatment showing the redistribution of both from IS to non-IS regions of the membrane. Representative quantifications are from three independent experiments of OTI T cells expanded from n = 3 OTI mice. **(K, N)** Localisation ratio of LCK and pLCK to the IS with vehicle (black dots) and Unc119i (red dots) treatment. Representative quantifications are from three independent experiments of OTI T cells expanded from n = 3 OTI mice, each dot representing an IS. All scale bars on confocal images represent 5 μm. All error bars represent the standard error of the mean. **(K, L, M, N)** Statistical significance on (K, L, M, N) was determined by the Mann–Whitney test.
Source data are available for this figure.

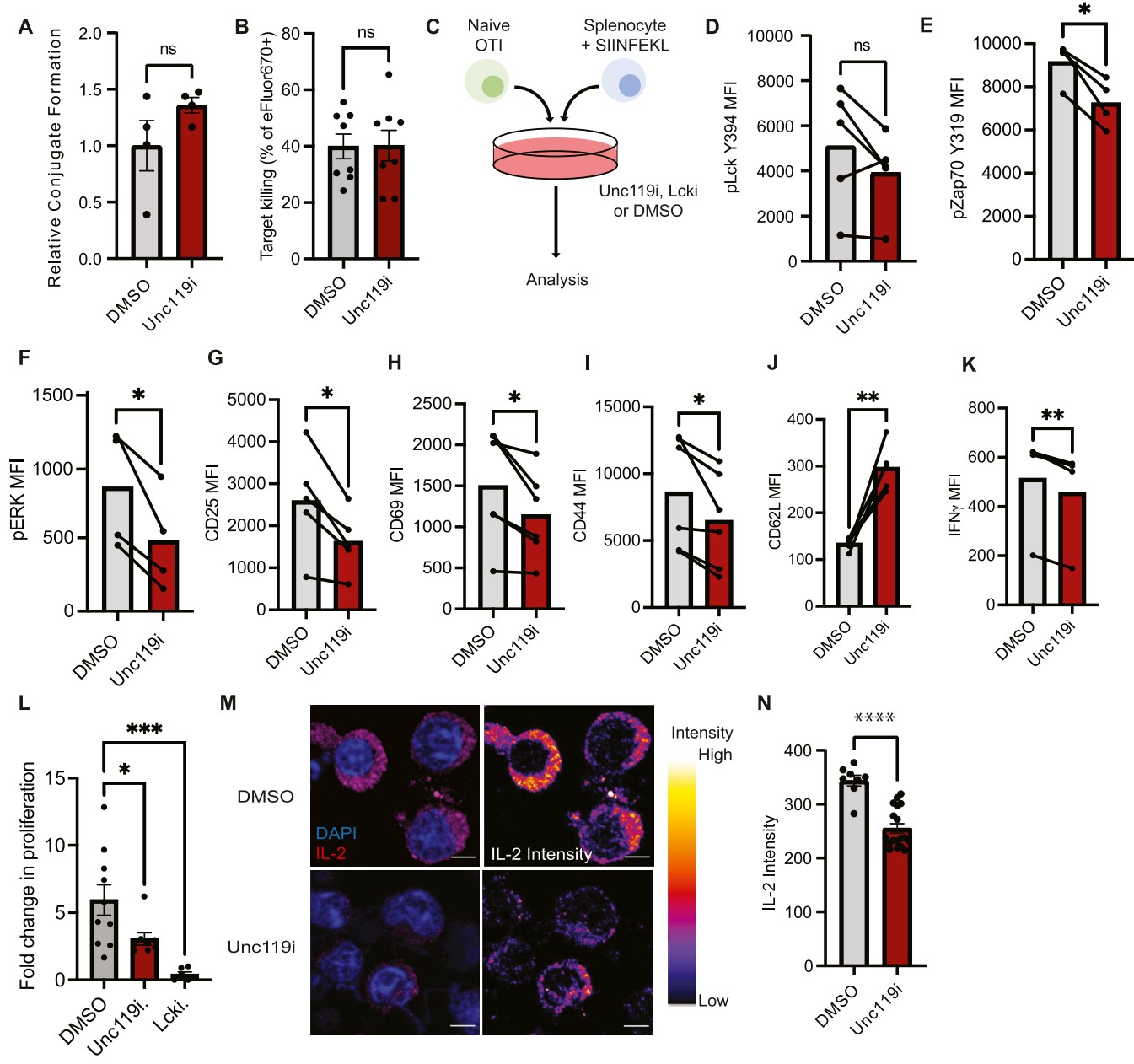

**Figure 3. Inhibition of Unc119 dampens CD8⁺ T-cell effector functions.**
**(A)** Flow cytometric quantification of in vitro–formed immune conjugates with vehicle- or Unc119i-treated OTI T cells. Quantifications are from two independent experiments. **(B)** Quantification of flow cytometry–based killing assay to assess the cytotoxic capacity of vehicle (DMSO)- and Unc119i-treated OTI T cells showing the percentage of dead target cells. Quantifications are from three independent experiments, each dot representing a technical replicate acquired after OTI T cells were isolated in vitro and plated individually in coculture with EG.7 target cells. **(C)** Schematic of an experiment to assess how Unc119i affects CD8⁺ T-cell effector functions. **(D, E, F, G, H, I)** Graphs showing the median fluorescence intensity of pLCK Y394, pZAP70 Y319, pERK, CD25, CD69, CD44, CD62L, and IFNγ, respectively, in CD8⁺ T cells after vehicle or Unc119i treatment. Data shown are from four to six independent experiments, each data point representing OTI cells isolated from one mouse. **(J, K)** CD62L and IFNγ median fluorescence intensity of OTI T cells activated in the presence of DMSO or UNC119i. Each dot represents technical replicates from two independent experiments. Statistical significance was determined by a paired *t* test. **(L)** Fold change in proliferation of CD8⁺ T cells with vehicle, Unc119i, or LCKi treatment. Each data point represents one biological replicate. Statistical significance was determined by one-way ANOVA. **(M)** Confocal imaging of intracellular IL-2 in CD8⁺ T cells with vehicle of Unc119i treatment. Representative images are from two independent experiments, each data point representing a cell. The scale bar represents 5 *μm*. **(M, N)** Quantification of IL-2 intensities shown in (M) done on ImageJ. All error bars represent the standard error of the mean.

facilitate the transcription of genes involved in T-cell activation. Using the same coculture system, we observed a significant reduction in ERK phosphorylation at Y192 in the presence of Unc119i (Fig 3F). Markers of T-cell activation were subsequently examined to

determine whether this had a functional consequence for T-cell differentiation. Levels of CD25, the high-affinity receptor for interleukin-2 (Fig 3G), CD69, an early activation marker of T cells (Fig 3H), and CD44, another activation marker of T cells (Fig 3I), were all

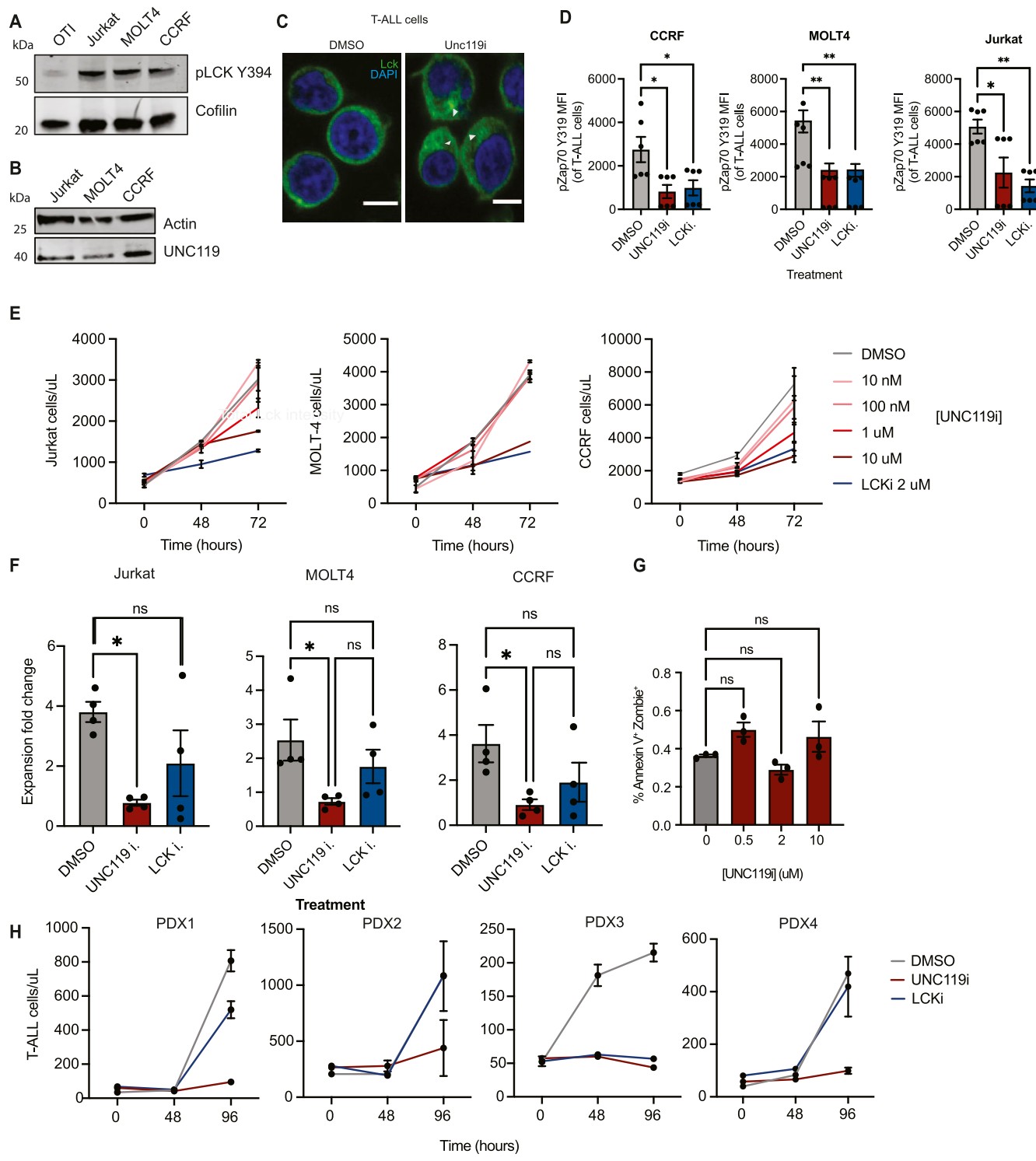

**Figure 4.  UNC119 inhibition reduces proliferation in T-ALL cell lines and PDX samples.**
**(A)** Western blot showing the expression of pLCK Y394 in healthy CD8[+] T cells (OTI) compared with Jurkat, MOLT4, and CCRF T-ALL cell lines relative to cofilin as a loading control. **(B)** Western blot showing the expression of UNC119 in Jurkat, MOLT4, and CCRF cell lines relative to actin as the internal loading control. **(C)** Confocal imaging of endogenous pLCK in T-ALL cells treated with either DMSO or Unc119i. White arrows on images indicate intracellular LCK stores. Representative images are from two independent experiments. Scale bars represent 5 $\mu m$. **(D)** Graphs showing the median fluorescence intensity of pZAP70 Y319 in human T-ALL cell lines with either DMSO, 5 $\mu m$ Unc119i, or 2 $\mu m$ LCKi (MedChemExpress—HY12072). Data shown are from two independent experiments, where each dot represents technical replicates from two experiments. Statistical significance was determined with one-way ANOVA on Prism GraphPad. **(E)** Proliferation curves of Jurkat, MOLT4, and CCRF cells with increasing concentrations of Unc119i and a fixed concentration of LCKi at 2 $\mu m$. Each condition and time point was plated in triplicate, each data point representing the average of three technical replicates. Error bars represent the standard error of the mean. One dose–response experiment was conducted per T-ALL cell line. **(F)** Graphs showing the

reduced in the presence of Unc119i, indicating that polarisation of LCK to the IS is critical for its normal function in naïve T-cell triggering. CD62L, a marker of naïve T cells, indeed remained elevated in the presence of Unc119i, confirming the reduced differentiation of T cells in the absence of LCK polarisation (Fig 3J). Finally, we showed that the production of IFNγ by activated CTLs was dampened in the presence of Unc119i (Figs 3K and S3A and B). Together, these results demonstrate that Unc119 is required for effective TCR signal transduction and for T-cell differentiation. Indeed, all markers examined showed similar changes as to when LCK itself was inhibited in OTI T cells, although to a lesser magnitude (Fig S4A–H). Given the requirement for T cells to expand upon antigen recognition to perform their function, we finally wanted to investigate whether Unc119 inhibition affected the expansion of T cells downstream of TCR. OTI cells were incubated with SIINFEKL-presenting splenocytes for 72 h, and then, their fold expansion was quantified by flow cytometry. In the presence of Unc119i, there was a significant reduction in fold expansion, indicating that the generation of an antigen-specific CTL pool depends on Unc119 (Fig 3L). Because autocrine IL-2 signalling is important for T-cell proliferation, especially in the in vitro context and the absence of CD4$^+$ T-cell help, we asked whether the defect in proliferation observed was due to impaired IL-2 production. Naïve OTI T cells were expanded in the presence of brefeldin A and imaged by confocal microscopy demonstrating strong IL-2 staining in the cytosol of vehicle-treated CTLs (Fig 3M), which was significantly reduced in Unc119i-treated CTLs (Fig 3M and N). When analysing these confocal images for an intensity profile, we observed pixels of high IL-2 intensity in vehicle-treated CTLs, which were largely absent in the presence of Unc119i (Fig 3M and N). Therefore, although needed for initiating TCR signalling in naïve T cells, UNC119 inhibition alone may not be sufficient to dampen cytotoxicity of existing effector OTI T cells in vitro, whereas direct inhibition of LCK has been shown to significantly reduce target cell killing (39).

## UNC119 inhibition represents a target in patient-derived T-ALL as an added strategy for dampening proliferation

Our data suggested that UNC119 inhibition impairs de novo priming but does not inhibit CTL function. Thus, if targeted in vivo, it may display a less severe immunosuppression than with LCK inhibition. To determine whether LCK signalling was also regulated by UNC119 in T-ALL and whether this could represent a therapeutic target, three different human T-ALL cell lines were examined. To quantify the level of phosphorylated LCK within T-ALL cells, whole-cell lysates of T-ALL cells and primary OTI T cells were subjected to Western blot analysis, which confirmed high levels of Y394-phosphorylated LCK in T-ALL cell lines (Figs 4A and S5A). UNC119, previously shown to be expressed in Jurkat cells (20), was also

expressed in the two additional human T-ALL lines examined (Figs 4B and S5B). Similar to primary CD8$^+$ T cells, UNC119 inhibition in human T-ALL cells was accompanied by a redistribution of LCK from being largely membrane-associated to also accumulating within the cytosol upon UNC119i treatment (Fig 4C). Because Unc119 inhibition in OTI T cells reduced pZAP70 levels, T-ALL cell lines were grown in the presence of DMSO, Unc119i, or Lcki and pZAP70 levels were quantified at the single-cell level using flow cytometry. This also revealed the decreased levels of pZAP70 median fluorescence intensity in UNC119i- and LCKi-treated T-ALL cells as compared to vehicle-treated cells (Fig 4D). The alteration in LCK localisation in addition to the reduced levels of pZAP70 in T-ALL cells with UNC119 inhibition resulted in a dose-dependent reduction in cell proliferation in all three cell lines (Fig 4E), with the most pronounced effect observed in CCRF T-ALL cells. To determine whether this reduced number of cells at 72 h was due to increased death or reduced proliferation, cells were stained for the presence of Ki-67, a marker of proliferation. All three cell lines showed a significant reduction in Ki-67 staining when treated with UNC119i or LCKi (Fig 4F). To determine whether there was also an increased amount of death in the presence of UNC119i, cells were stained for Annexin V after 48 h of culture with increasing concentrations of UNC119i. At 48 h, there was only a small proportion of apoptotic cells and there was no effect of UNC119i on cell death (Fig 4G). Taken together, these experiments demonstrate that UNC119i may represent a therapeutic target to reduce T-ALL proliferation. To determine whether this observation applied in primary patient-derived samples, we repeated the in vitro coculture experiments of T-ALL cells from patient-derived xenografts with stromal cells in the presence of UNC119i or LCKi. Because T-ALL cells grow in suspension and stromal cells are adherent, this allowed for an easy separation of leukaemic cells from stromal cells. Reassuringly, all PDX samples showed a good reduction in proliferation with UNC119i and LCKi inhibitor treatment (Fig 4H).

## Genetic depletion of UNC119 in human T-ALL cell lines phenocopies defective proliferation

As UNC119i showed a strong effect in some cell lines and patient samples, we wanted to confirm that this was not due to off-target effects of squarunkin A. To that end, we generated a CCRF cell line expressing a fluorescent Cas9 and a doxycycline-inducible fluorescent gRNA targeting UNC119 because the strongest effect with the inhibitor was observed in CCRF cells. We then conducted a time course experiment quantifying UNC119 levels by Western blot analysis at various time points post-doxycycline treatment. This showed a pronounced decrease in the amount of UNC119 at 48 and 96 h post-treatment with doxycycline confirming the knockdown was functional (Figs 5A and S5C). We subsequently characterised

fold change in proliferation at endpoint of the same three human T-ALL cell lines. Four independent experiments were conducted. Each condition was plated in triplicate, and fold change was calculated by dividing the average cell concentration at 96 h by that at 0 h. Statistical significance was determined with one-way ANOVA on Prism GraphPad. **(G)** Graph showing the percentage of Annexin V+ Zombie NIR+ apoptotic T-ALL cells with increasing concentrations of Unc119 inhibitor. **(H)** Proliferation curves of four different PDX-derived human T-ALL cells in response to Unc119i and LCKi treatments. Each dot is the average of triplicate plating for each PDX sample. All error bars represent the standard error of the mean.
Source data are available for this figure.

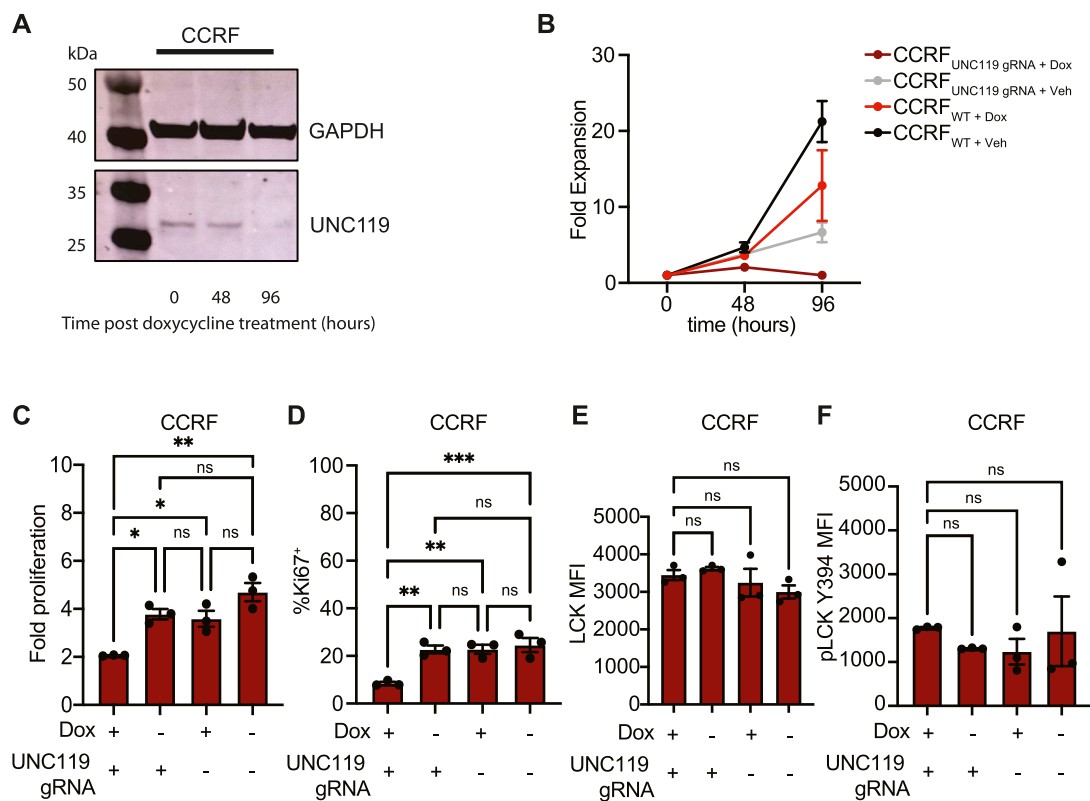

**Figure 5. Genetic depletion of UNC119 in human T-ALL cell lines phenocopies defective proliferation.**
**(A)** Western blot showing depletion of UNC119 in CCRF cells that were engineered to express a doxycycline-inducible gRNA targeting UNC119, at two different time points post-doxycycline treatment. GAPDH was the internal loading control. **(B)** Growth curve of UNC119 WT and KO (CCRF$_{Unc119\ gRNA\ +\ DOX}$) cells over time. Data are representative of two independent experiments; each dot on the graph shows an average of three technical replicates plated per condition. **(C)** Graph showing fold change in proliferation of modified CCRF cells. Data are representative of two independent experiments where the graph shows three technical replicates from one representative experiment. **(D)** Percentage of Ki-67+ CCRF cells that are either sufficient or deficient for UNC119 as quantified by flow cytometry. **(E, F)** MFI of total LCK and pLCK Y394, respectively, in UNC119-sufficient (Dox+ UNC119 gRNA+) and UNC119-deficient CCRF cells (all other conditions). Data are representative of two independent experiments. All error bars represent the standard error of the mean. Statistical significance was determined using one-way ANOVA on Prism GraphPad.
Source data are available for this figure.

the effect of UNC119 depletion in T-ALL cells in vitro using UNC119-depleted CCRF-CEM cells as a model. By expanding cells with or without inducible *UNC119* gRNA in the presence or absence of doxycycline for 96 h, we saw that CCRF$_{UNC119\ gRNA\ +\ (DOX)}$ cells expanded the least as expected (Fig 5B and C). Flow cytometric analysis of intracellular Ki-67 also showed that CCRF$_{UNC119\ gRNA\ (+DOX)}$ cells had a significantly lower expression of Ki-67 indicating reduced proliferation (Fig 5D). Once again, we observed no change in the levels of either total LCK (Fig 5E) or its active Y394-phosphorylated form (Fig 5F), indicating that whether UNC119 was depleted or not, this did not affect activation of LCK, in which case these cells phenocopied primary CTLs with UNC119 inhibition.

## Discussion

The work presented here highlights that organisation at the intracellular face of the IS is critical for full TCR signal transduction in mature T cells. In the absence of UNC119-mediated shuttling of LCK to the IS, there is reduced TCR signal transduction as shown by

impaired phosphorylation of ZAP70, poor expression of activation markers, and defective proliferation of OTI T cells. Because UNC119 inhibition did not significantly reduce the amount of Y394 activating phosphorylation of LCK, we suggested there exists a decoupling between the IS-specific localisation of LCK and its activating phosphorylation and demonstrated the requirement in CD8+ T cells for IS localisation of LCK to activate ZAP70. As such, TCR triggering relies upon redistribution of pLCK and not simply changes in phosphorylation or kinetic segregation of phosphatases or endosomal shuttling of LCK towards the IS. Indeed, impairment of de novo T-cell activation with UNC119 inhibition may relate to the requirement for LCK to relocalise to form molecular bridges with ZAP70 (40). Critically, this work also highlights that despite squarunkin A only binding to the lipid binding cleft in UNC119, this was sufficient to prevent LCK localisation. This, and previously published work in Jurkat cells (20), suggests that the major mechanism by which UNC119 acts is by solubilising LCK and trafficking it to the IS as opposed to through activating RAB11-dependent endosomal traffic (13).

This work also helps explain the lack of both circulating CD4+ and CD8+ T lymphocytes in patients harbouring a dominant negative

mutation in UNC119 (22). In vitro inhibition of UNC119 in isolated T cells leads to poor activation of LCK and downstream targets, suggesting that this deficit in T cells may be directly a result of their inability to relocalise LCK to the plasma membrane. It would be interesting to see whether this extends to T-cell development within the thymus and whether the reduction in T cells in these patients is due to failure to generate T cells initially within the thymus during positive selection or a reduction in homeostatic proliferation and maintenance. Furthermore, it would be of interest to determine whether the alteration in LCK activation changed the range of TCR affinities possessed by T cells egressing the thymus. Indeed, these patients also present with defective T-cell proliferation, which was recapitulated in our results in CD8+ T cells that showed similar defective proliferation when UNC119 was inhibited.

A surprising observation was that UNC119 inhibition does not lead to dampening of CTL killing responses even though the polarisation of pLCK was reduced in the presence of Unc119i, whereas LCK inhibition did dampen cytotoxicity and mirrored phenotypes observed with the use of dasatinib (41) and when LCK was genetically depleted in primary CD8+ T cells (39). This may in part be due to the highly robust activation of the OTI TCR by the SIINFEKL peptide, which possibly overrides defects in killing induced by Unc119i. It would be interesting to determine whether more normal, physiological CD8+ T-cell responses are also refractory to Unc119i in future. It is also possible, however, that the reduced sensitivity is because although UNC119i reduced polarisation, it did not totally remove pLCK from the IS. A fraction of LCK within T cells is CD4 and CD8 coreceptor–bound (42), and because UNC119 inhibition did not completely redistribute LCK from the plasma membrane to the cytosol, and does not dampen the activating phosphorylation of LCK, it is possible that the remaining LCK on the plasma membrane is sufficient to drive centrosomal docking and subsequent cytotoxic killing of target cells. It is also a possibility that in the absence of LCK function, FYN signalling, which is reported to promote stronger proliferation and survival of CD8+ T cells, compensates to drive cytotoxicity, as has been shown in chimeric antigen receptor T-cell models (43). Furthermore, CD8-associated LCK has been shown to be required for responses to suboptimal antigen while largely dispensable for cytotoxic function, largely mirroring our findings that LCK inhibition affects primary signalling more than effector function (44). Interestingly, in this study it was also seen that CD4-bound LCK was required for effective development and function of CD4 T cells. As such, it would be of interest to determine whether UNC119i has a lesser role in CD4 T cells where coreceptor-bound LCK appears to be more important for normal TCR signalling (44).

The significance of the UNC119 trafficking of LCK is further highlighted by the phenotypes observed in patient-derived samples where UNC119 inhibition reduced ZAP70 phosphorylation and greatly suppressed proliferation. Although LCK has been extensively studied as a target in T-ALL, UNC119 represents a new potential target through which to modulate LCK activity. This is important not only for the cytostatic effect observed but also because aberrant LCK signalling confers resistance to some therapeutic interventions including glucocorticoid (GC) treatment in paediatric T-ALL (35, 45). It has also been reported that LCK overexpression in T-ALL leads to the aberrant activation of NFAT, and the production of excessive IL-4 enabling evasion of GC-mediated apoptosis (35). Because GC treatment is administered to acute lymphocytic leukaemia patients at various stages of therapy, a greater understanding of the mechanisms underlying resistance in prednisone-poor responders is needed. LCK itself is being investigated in studies aiming to target it for degradation by the proteasome through tagging with PROTAC (46), but UNC119 may provide an alternative strategy, which could preserve the capacity of existing cytotoxic responses.

Although not all T-ALL cell lines express a membrane-bound TCR (such as the MOLT-4 cell line), LCK signalling still occurs in T-ALL cells. In T-ALL, the SRC kinases LCK and FYN have been reported to form a complex with heat shock protein 90 and glucocorticoid receptor (GR), which when ligated with GC promotes the dissociation of the complex and impaired TCR signalling (47). This treatment strategy is often unsuccessful because of the overactivation of LCK in T-ALL cells coupled with the mutation of GR found in some patients with T-ALL. Our data show that inhibiting UNC119 in T-ALL cells reduced proliferation, which suggests T-ALL cells may still require UNC119 to correctly localise LCK and drive downstream signalling via activation of calcineurin and NFAT, which are indispensable for T-ALL survival and expansion (48). This dependence on UNC119 for proliferation has previously been shown in colorectal cancer cell lines (49), which rely on LCK signalling considerably less than T-ALL cells do. As such, UNC119 may represent a novel therapeutic target in T-ALL, which has the potential to produce less immunosuppression than LCK targeting. This is supported by the fact that inhibition of UNC119 had a greater effect on the in vitro proliferation of some patient-derived cells. This may be due to the relative efficacy of the inhibitors used in this study or may indicate that inhibition of UNC119 has other, secondary, effects. However, it suggests that UNC119 inhibition may expand the scope of patients who could benefit from therapies targeting LCK. This will require the generation of new strategies to target UNC119 as squarunkin A has poor pharmacodynamic properties making it unsuitable for in vivo use. Further work will also be needed to determine the impact of inhibiting UNC119 on ciliated cells throughout the rest of the body, and if these are problematic, drug conjugates targeting the inhibitors to lymphocytes may be required.

Several aspects of this signalling pathway remain unanswered; however, our work further highlights that ciliary proteins ought to be studied more in the anti-cancer immune response, and this is twofold. Firstly, we do not know the effect of depleting ARL3 and ARL13b in the T-ALL setting. Secondly, it would be valuable to understand the transcriptomic changes of both healthy T cells and T-ALL cells in the context of UNC119 depletion with RNA sequencing, to better characterise the transcriptomic changes effected by UNC119. Both approaches could identify novel targets in T-ALL.

# Materials and Methods

### Animal work

All animal work was in accordance with the animal ethics and welfare committee at the University of Glasgow and U.K. Home Office regulations (ASPA, 1986, PPL P72BA642F). All mice were bred and housed at the Cancer Research Scotland Institute. C57BL/6J

mice were purchased from Charles River Laboratories (United Kingdom). OTI mice were purchased from the Jackson Laboratory (United States).

## scRNAseq analysis

Data were obtained from the Gene Expression Omnibus with accession codes GSE126030 (human) and GSE130130 (mouse). Samples were combined, and an integrated analysis was run using Seurat. Clusters were determined using k-means clustering, and dimensionality reduction was carried out using UMAP. RNA levels were compared using violin plots.

## UNC119: squarunkin A complex structure determination

UNC119 was purified as previously described (50). Before sparse matrix screening, the protein was incubated with a twofold molar excess of squarunkin A. Crystals were obtained in the JBS Kinase screen in a condition containing 30% wt/vol PEG3350, 100 mM sodium acetate, pH 4.6, and 200 mM ammonium acetate. Data processing was performed using the Xia2 pipeline. Molecular replacement was performed using Phaser (51) with a single chain from PDB: 6H6A (20) as a search model. Refinement was performed using REFMAC5 (52) and COOT (53) of the CCP4 program suite (54). Squarunkin library dictionary was performed using eLBOW from Phenix.refine (55).

## CRISPR constructs

The CRISPR system was made as previously described (56). A gRNA for human *UNC119* was cloned into a vector to obtain the inducible expression of guide RNA with a fluorescent GFP reporter (plasmid #70183, FgH1tUTG; Addgene). The Cas9 mCherry vector used was FUCas9Cherry (plasmid #70182; Addgene). The gRNA sequence for human *UNC119* was /5Phos/TCCCCAGAACGGTTGCCCATCAAC (Integrated DNA Technologies) and was cloned into plasmid #70183 with BsmBI restriction sites.

## Production of virus and transduction of cell lines

The HEK293FT cell line was used as the packaging cell line to produce lentiviral particles. 293FT cells were transiently transfected in 10-cm Petri dishes, with 10 $\mu$g of vector DNA, and associated packaging constructs: 5 $\mu$g pMDL, 2.5 $\mu$g pRSV-rev, and 3 $\mu$g pVSV-G (Addgene), and transfected using calcium phosphate precipitation. Cell supernatants containing virus were harvested 72 h post-transfection and filtered through a 0.2-$\mu$m filter. Target leukaemia cells were transduced with virus using spinfection. Targets were plated at 500,000 cells/ml in a 24 well in 1 ml of viral supernatant and 1 ml of DMEM with 10% heat-inactivated FCS and 10 ng/$\mu$l polybrene per well. Plates were centrifuged at 865$g$ for 2 h at 32°C. Post-centrifugation, cells were resuspended in the well and incubated for 24 h at 37°C.

## FACS of transduced cell lines

Cells were sorted for either mCherry positivity, GFP positivity, or both on the BD FACSAria sorter. Cells were stained with DAPI as a viability marker diluted at 1:500 in PBS. Cells were stained for 25 min at 4°C and washed with PBS by centrifugation, followed by

resuspension in PBS/2% FBS before acquiring on the Aria sorter. Recovery media used were complete RPMI supplemented with 20% heat-inactivated FBS.

## Cell culture

Human cell lines used in this study include Jurkat, MOLT-4, and CCRF-CEM, originally purchased from the ATCC and regularly authenticated through STR profiling and mycoplasma testing. Cells were cultured in DMEM, supplemented with 2.5 mM L-glutamine, 1% penicillin–streptomycin, and 10% heat-inactivated FCS. Murine cell lines used in this study were EG.7, originally purchased from ATCC and regularly authenticated through STR profiling and mycoplasma testing. Cells were cultured in RPMI, supplemented with L-glutamine, 1% penicillin–streptomycin, and 10% heat-inactivated FCS.

## Cytotoxicity assays

Flow cytometry–based cytotoxicity assays were performed to assess differences in OTI-mediated killing of target cells with Unc119 inhibition. OTI effector T cells were cultured in RPMI supplemented with L-glutamine, 1% penicillin–streptomycin, 10% heat-inactivated FCS, 10 U/ml recombinant IL-2, 50 $\mu$m beta-mercaptoethanol, and either DMSO or Unc119 inhibitor (Unc119i) at 5 $\mu$m. OTI cells were then added to eFluor670-labelled target EG.7 lymphoma cells at an effector: target ratio of 5:1. Cocultures were incubated at 37°C for 2 h and then acquired by flow cytometry.

## Drug treatments

### UNC119 inhibition
Squarunkin A hydrochloride was purchased from Bio-Techne (Catalog #6364), and used at 5 $\mu$m, dissolved in DMSO, and added to cells while in complete medium.

### LCK inhibition
LCK inhibitor ((6-(2,6-dimethylphenyl)-2-((4-(4-methylpiperazin-1-yl) phenyl)amino)benzo[4,5]imidazo[1,2-a]pyrimido[5,4-e]pyrimidin-5(6H)-one)) was purchased from MedChemExpress (Catalog #HY-12072), and used at a working concentration of 2 $\mu$m, dissolved in DMSO, and added to cells while in complete medium. LCK and squarunkin A inhibitors were prepared at a stock concentration of 20 and 50 mM, respectively. The DMSO dilution per well in cell culture was 1:2,000.

### Doxycycline-mediated gRNA induction
Doxycycline was purchased from Merck (Catalog #D9891) and used at a working concentration of 8 $\mu$g/ml.

## Patient-derived xenograft samples

T-ALL primary cells to generate patient-derived xenograft mice were obtained from Dr Sandra Marmioli and made in collaboration with the oncohaematology laboratory in Padova. Written informed consent for the use of leftover material for research purposes was obtained from all patients at the trial entry of the Associazione Italiana di Ematologia ed Oncologia Pediatrica, and the Berlin-Frankfurt-Münster (AIEOP-BFM) ALL 2000/2006 paediatric clinical

protocols in accordance with the Declaration of Helsinki. PDX samples were cultured for the duration of the proliferation experiment with HS5 stromal cells at a ratio of 1:1, in RPMI medium supplemented with 10% heat-inactivated FBS, 2.5 mM L-glutamine, and 1% penicillin–streptomycin.

## Generation of activated OTI T cells in vitro

OTI T-cell receptor transgenic mice were used in this study, and these were of the C57BL/6 strain, male in gender, and between 12 and 15 wk of age. Spleen and lymph nodes (LN) were dissected from OTI TCR transgenic mice, and single-cell suspensions were prepared by both mashing the organs through a 70-$\mu$m filter and washing with complete medium. Red blood cell lysis was performed on splenocytes and LN-derived cells, using a premade RBC lysis buffer (Cat #420302; BioLegend). Splenocytes were pulsed with ovalbumin peptide (SIINFEKL), for 45 min at 37°C, and washed with complete RPMI three times. LN-derived cells were then added to the splenocytes, and cells were cultured for 1 wk in RPMI supplemented with 10% heat-inactivated FCS, 1% penicillin–streptomycin, 2.5 mM L-glutamine, and 50 $\mu$m beta-mercaptoethanol. 10 U/ml of recombinant murine IL-2 was added to the expanding culture every 2 d.

## Western blotting

For SDS–PAGE, gels were placed in Bio-Rad TGX gel tanks filled with MOPS running buffer (NuPAGE MOPS SDS Running Buffer, Cat #NP000; Thermo Fisher Scientific). A prestained protein ladder was used to determine molecular weight (Cat #26616; Thermo Fisher Scientific). Bio-Rad precast 4–12% Bis-Tris gels were used to separate proteins, at a voltage of 150 V for 60 min. Gels were transferred onto a nitrocellulose membrane through wet transfer at 30 V for 70 min. Membranes were blocked in Rockland blocking buffer (#MB-070; Rockland Immunochemicals) for 1 h at room temperature. Primary antibodies were diluted in Rockland blocking buffer and incubated overnight at 4°C. Then, membranes were washed three times with TBS/Tween and incubated on a rolling shaker with a Li-COR secondary antibody for 1 h at room temperature. Li-COR secondary antibodies were either IRDye 800 or IRDye 680 purchased from Li-COR and diluted in blocking buffer at a dilution of 1:5,000. Membranes were acquired on the Odyssey instrument (Li-COR). Antibodies used were as follows: total LCK (ab3885; Abcam), phospho LCK Y394 (MAB7500; Biotechne) Vinculin (ab129002; Abcam), ARL13B (ab83879; Abcam), ARL3 (10961-1-AP; ProteinTech), UNC119 (13065-1-AP; ProteinTech), GAPDH (MA5-15738; Invitrogen), Actin (MA1-744; Invitrogen), or Cofilin (MA5-17275).

## Flow cytometry staining

The flow cytometer used for all experiments was the BD LSRFortessa. Cells were harvested in a V-bottom 96-well plate and centrifuged for 5 min at 435$g$. Cells were then stained with a viability marker (Zombie NIR, Biolegend or DAPI) diluted in PBS for 20 min at 4°C. Extracellular staining antibodies were diluted in PBS and incubated for 20 min at 4°C. Cells were then washed and either acquired directly or fixed in fixation buffer (Cat #420801; BioLegend),

and kept at 4°C until acquisition by flow cytometry. For intracellular staining, samples were permeabilised after fixation, with intracellular permeabilisation buffer (Cat #421002; BioLegend) by resuspending cells in buffer and centrifugation for 10 min at 350$g$ at 4°C. Antibodies were diluted in permeabilisation buffer and incubated for 25 min at 4°C. Antibodies used were as follows: CD8 alpha (PerCP-Cy5.5, mouse, Cat #104721; BioLegend), CD3 (APC, mouse, Cat #100202; BioLegend), Va5.1 T-cell receptor (PE, mouse, Cat #139503; BioLegend), CD69 (BV650, mouse, Cat #104541; BioLegend), CD25 (FITC, mouse, Cat #101907; BioLegend), CD44 (BV711, mouse, Cat #103057; BioLegend), CD69 (APC, mouse, Cat #MHCD6905; Thermo Fisher Scientific), phosphorylated LCK Tyr 394 (PE, mouse, Cat #933104; BioLegend), phosphorylated ZAP70 (Tyr 319, PE, mouse, Cat #683702; BioLegend).

## Annexin V apoptosis assay

The apoptosis assay kit was purchased from BD Biosciences (Cat #556421), and followed according to the manufacturer's protocol.

## In vitro immune conjugate formation

13-mm coverslips were coated with 10 $\mu$g/ml fibronectin diluted in PBS, at 37°C for 1 h. OTI T cells and EG.7 lymphoma cells (purchased from the ATCC) were coincubated at a ratio of 1:1 at a concentration of 2,000,000 cells/ml for 20 min on fibronectin-coated coverslips. At time point end, conjugates were visualised under the microscope to check for adherence, and then fixed in 4% PFA for 15 min at room temperature. EG.7 cells were labelled with CellTrace Violet to allow for easy separation of effector and target cells.

## Immunofluorescence staining

Fixed OTI and EG.7 immune conjugates were fixed using 4% PFA and permeabilised using 0.5% Triton X-100 to allow for the visualisation of intracellular proteins. Coverslips were then incubated in DAKO blocking solution (Agilent Technologies) for 1 h, and antibodies were used at a dilution of 1:200 and diluted in blocking solution. Coverslips were incubated overnight at 4°C. Antibodies used for IF were against total LCK, Y394-phosphorylated LCK, gamma-tubulin, ARL3, UNC119, RAB11, or CD8 depending on the experiment. Coverslips were washed three times in PBS before the addition of secondary antibodies (diluted 1:500 in blocking solution). The secondary antibody was incubated for 1 h at room temperature in the dark. The coverslips were washed three times in PBS before mounting them onto glass slides on a ProLong Gold mountant. Slides were left to set for at least 24 h before imaging. Antibodies used were as follows: phosphorylated LCK Tyr 394 (Cat #PA5-39696; Thermo Fisher Scientific), total LCK (Cat #Ab3885; Abcam), CD8$\alpha$ alpha (Cat #NBP1-49045SS; Novus Biologicals), UNC119 (Cat #13065-1-AP; ProteinTech), ARL13B (Cat #Ab83879; Abcam), and ARL3 (Cat #10961-1-AP; ProteinTech).

## Confocal imaging and image analysis

Optical sectioning of fixed conjugate cells was performed using the confocal laser scanning microscope (LSM) 710 at 63X. Z-stacks were

obtained to cover the entire IS. Images were viewed and analysed using ImageJ (49), and localisation ratios of protein at the IS were quantified using SynapseMeasure, to obtain a measure of how much a protein is polarised to the IS relative to the rest of the T-cell membrane as previously described (57).

### Localisation ratio analysis of immune conjugates—ImageJ

A plugin called SynapseMeasure on ImageJ was used to analyse how much a protein was polarised to the IS relative to the rest of the T-cell membrane. Regions of interest were obtained for background, target cell membrane, T-cell membrane, and IS membrane (schematic, Fig 2H). The background intensity was subtracted from each region of interest, the intensity coming from the target cell was subtracted from the IS intensity, and this value was then divided by the intensity of the T-cell membrane, which results in a localisation ratio of protein at IS:T-cell membrane (49).

### Profile plot analysis—ImageJ

The profile plot function on ImageJ was used to measure the intensity of a protein across a cell going from one side of the membrane to the other via the cytosol. A line segment was drawn across the cell, and the graph generated shows how the intensity of a protein changes across the length of the line, thereby giving information on subcellular localisation and protein intensity.

### Protein intensity analysis—ImageJ

To measure the intensity of protein inside the cell, two intensity values were measured: the intensity of the whole cell (cell border + cytosol). The intensity of each region was determined by drawing a line segment around the area. Intracellular protein intensity was calculated by subtracting cell border intensity from whole-cell intensity.

### Statistics

GraphPad Prism version 9 was used to plot data and assess statistical differences of the data. Statistical significances were determined by a two-tailed unpaired *t* test or paired *t* test for experimental analysis of different groups from the same animals, and two-way ANOVA or 1-way ANOVA for multiple group comparisons.

## Data Availability

X-ray crystal structure data are available in the PDB database with the accession number 9GKG.

## Supplementary Information

## Acknowledgements

We thank the flow cytometry core facility at the CRUK Scotland Institute for FACS cell lines and the Beatson Advanced Imaging Resource for their assistance with confocal microscopes. The article was critically reviewed by Catherine Winchester (CRUK Scotland Institute). This work was supported by CRUK core funding to the Scotland Institute (A31287), CRUK core funding to EW Roberts (A1920), CRUK core funding to PA Gammage (A1920), European Research Council (ERC) Starting Grant to PAM (EP/X035581/1), NIH grant to PA Gammage (R37CA276200), CCLG Project Grant to C Halsey (CCLGA 2020 24), CRUK Programme Foundation Award to C Halsey (DRCPFA-Nov21\100001), Associazione Italiana per la Ricerca sul Cancro Individual Grant 2016 to S Marmiroli (19186), Not-for-profit Associazione Mantovana per la Ricerca sul Cancro to S Marmiroli (2021/E95F21001040007), Not-for-profit Associazione Mantovana per la Ricerca sul Cancro to S Marmiroli (2023/E93C23000570007), and Associazione Italiana per la Ricerca sul Cancro under MFAG 2018 to V Serafin (21771).

## Author Contributions

Y Samarakoon: conceptualisation, formal analysis, investigation, visualisation, and writing—original draft, review, and editing.
T Yelland: investigation.
E Garcia-Gonzalez: investigation.
A da Silva Justo Junior: investigation.
M Mahmood: investigation.
A Manoharan: investigation.
S Patterson: investigation.
V Serafin: investigation.
PA Gammage: funding acquisition and validation.
S Marmiroli: investigation.
C Halsey: investigation.
S Ismail: conceptualisation, formal analysis, supervision, funding acquisition, investigation, visualisation, and writing—original draft, review, and editing.
EW Roberts: conceptualisation, formal analysis, supervision, funding acquisition, investigation, visualisation, project administration, and writing—original draft, review, and editing.

### Conflict of Interest Statement

The authors declare that they have no conflict of interest.

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
