## [Reviewer comments · Life Science Alliance]

Life Science Alliance

UNC119 regulates T cell receptor signalling in primary T cells and T acute lymphocytic leukaemia

Youhani Samarakoon, Tamas Yelland, Esther Garcia-Gonzalez, Amauri Justo-Junior, Mahnoor Mahmood, Anand Manoharan, Shaun Patterson, Valentina Serafin, Payam Gammage, Sandra Marmioli, Christina Halsey, Shehab Ismail, and Edward Roberts

DOI: <https://doi.org/10.26508/lsa.202403066>

Corresponding author(s): Edward Roberts, Cancer Research UK Scotland Institute; Youhani Samarakoon, The Beatson Institute for Cancer Research; and Shehab Ismail, KU Leuven

Review Timeline:

Submission Date:	2024-09-27
Editorial Decision:	2024-11-11
Revision Received:	2024-12-16
Editorial Decision:	2024-12-19
Revision Received:	2024-12-24
Accepted:	2024-12-26

Transaction Report:

November 11, 2024

Re: Life Science Alliance manuscript #LSA-2024-03066-T

Dr. Ed W Roberts
Cancer Research UK Beatson Institute
Switchback Road
Glasgow, Scotland G61 1BD
UNITED KINGDOM

Dear Dr. Roberts,

Thank you for submitting your manuscript entitled "UNC119 regulates cytotoxic and malignant T cell responses in TCR transgenic models and human T acute lymphocytic leukaemia" to Life Science Alliance. The manuscript was assessed by expert reviewers, whose comments are appended to this letter. We invite you to submit a revised manuscript addressing the Reviewer comments.

Thank you for this interesting contribution to Life Science Alliance. We are looking forward to receiving your revised manuscript.

Sincerely,

B. MANUSCRIPT ORGANIZATION AND FORMATTING:

Reviewer #1 (Comments to the Authors (Required)):

The Src family tyrosine kinase Lck is a key regulator of proximal T cell receptor signaling. In the activation of T cells by antigen presenting cells, recruitment of Lck to the cellular interface is required for effective function. As established in CD4 T cells, trafficking of Lck from the Golgi to the cellular interface is enhanced by the myristoyl binding protein UNC119. Physiological significance is supported by association of UNC119 mutations with CD4 T lymphopenia and targeting of UNC119 by HIV Nef. Here Samarakoon et al extend investigations of the role of UNC119 to CD8 T cells and T acute lymphocytic leukemia (T-ALL). In a reanalysis of published scRNAseq data of murine and human CD8 T cells, UNC119 and its interactors ARL3 and ARL13B are shown to be widely expressed. A co-crystal structure of UNC119 with an established inhibitor, Sqarunkin A, provides clear evidence of competition for UNC119 binding by Lck versus the inhibitor. Using an antigen-driven mouse model of CD8 T cell activation, the authors confirm that Unc119 inhibition impairs recruitment of Lck to the interface between an activating T cell and an antigen presenting cell. Cytolytic function of murine CD8 cytotoxic T lymphocyte function was not impaired. However, in the activation of naïve CD8 T cells, Unc119 inhibition led to multiple defects in proximal T cell signaling, proliferation and cytokine secretion. Suggesting a potential therapeutic application of these findings, the authors show that proliferation of three T-ALL cell lines and patient-derived cells was similarly reduced by UNC119 inhibition with comparably impaired Lck localization. The UNC119 inhibitor data were corroborated by UNC119 knock down in one of the T-ALL cell lines. In combination, these data constitute an interesting extension of insight into UNC119-mediated Lck trafficking to CD8 T cells with potential therapeutic application in T-ALL. There are only a number of minor concerns.

1. The authors use Fig. 3D and E to argue that while Lck activation as measured by phosphorylation of Lck at Y394 is not altered by Unc119 inhibition, activation of the downstream Lck target ZAP-70 is. This interpretation is based on differences between control and treated cells being significant for ZAP-70 but not phosphorylated Lck. However, both assays show a comparable reduction in the mean of phosphorylation. The phospho-Lck data just happen to be a bit more variable. A quick test shows that doubling the size of the phospho-Lck data would almost certainly yield a significant difference. While repeating the phospho-Lck assay of Fig. 3D is likely beyond the scope of a reasonable review request, the authors should edit the way they interpret their data in acknowledging that it is uncertain whether Lck phosphorylation is reduced or not, the change in mean pointing in one direction, the analysis of statistical significance (weakly) in the other.
2. The authors should quantify band intensities relative to control in Fig. 4A and B, in particular as gel loading seems uneven. Residual expression of UNC119 in Fig. 5A should also be quantified. The Western blot suggests that the authors have achieved a knock down, not a knockout. This doesn't affect the validity of their conclusions but should nevertheless be acknowledged in the description of the data.
3. It would be good to show all single cell data across the three experimental replicates in Fig. 2K and L (rather than representative data). This can effectively be done using 'SuperPlots' (<https://doi.org/10.1083/jcb.202001064>).
4. The figure caption for Fig. 4 list a panel I (that should be useful) - such panel is not part of the figure though.

Reviewer #2 (Comments to the Authors (Required)):

The authors demonstrated that despite lacking cilia, the ciliary proteins UNC119, ARL13b, and ARL3 are expressed widely in human T cell subsets, suggesting roles across various T cell stages. Inhibition of UNC119 in naïve CD8 T cells reduces LCK polarization and TCR signaling, though it does not hinder cytotoxic function in activated T cells. Additionally, UNC119 inhibition in T-ALL cells leads to reduced ZAP70 phosphorylation, decreased cell proliferation, and suppressed growth in patient-derived T-ALL samples, suggesting UNC119 as a candidate therapeutic target for T-ALL.

Overall, this study is well-executed and presented with clarity. However, I have a few comments for the authors to consider:

1. The term "transgenic models" in the title may be misleading, as it typically implies genetically-engineered disease models, which is not the case here. Please consider using a different term for clarity.
2. Please review line 136, as there is an inserted commentary.
3. In line 138, please clarify the distinction between UNC119 and UNC119A.
4. Regarding lines 322-323, you state that "UNC119 has a functional role in all subsets of T cells and across different stages of

differentiation." Did the single-cell dataset include only mature T cells, or did it also cover precursor/immature cells such as thymic seeding precursor cells and DN cells, representing the full T cell differentiation trajectory? Please clarify this throughout the manuscript.

5. Harmonize the y-axis scales in Figures 1B/C and 1E/F, as inconsistent scales may mislead readers.

6. In Figure 4, specify the LCK inhibitor used; if it was dasatinib, the concentration appears high, given its reported efficacy at nanomolar levels (10 to 100 nM) in the literature. Please discuss this and note that the UNC119 inhibitor seems markedly more effective than the LCK inhibitor in reducing cell proliferation, which is perhaps worth highlighting.

7. Have you examined the intracellular distribution of UNC119 following inhibitor treatment? Considering the pronounced proliferation inhibition observed, it would be insightful to explore potential mechanisms, particularly as T-ALL cells without TCR engagement do not form an immunological synapse.

We would like to thank the reviewers for their insightful and helpful comments. It's a busy time for all of us so it is much appreciated that they took time to review our manuscript. We have made several changes in line with the comments received including softening some language where appropriate, adding in some quantification, using superplots and textual amendments. We believe this has improved the manuscript and addressed the concerns raised. Below is a point by point reply to reviewers' comments to highlight these changes:

Reviewer #1

1. The authors use Fig. 3D and E to argue that while Lck activation as measured by phosphorylation of Lck at Y394 is not altered by Unc119 inhibition, activation of the downstream Lck target ZAP-70 is. This interpretation is based on differences between control and treated cells being significant for ZAP-70 but not phosphorylated Lck. However, both assays show a comparable reduction in the mean of phosphorylation. The phospho-Lck data just happen to be a bit more variable. A quick test shows that doubling the size of the phospho-Lck data would almost certainly yield a significant difference. While repeating the phospho-Lck assay of Fig. 3D is likely beyond the scope of a reasonable review request, the authors should edit the way they interpret their data in acknowledging that it is uncertain whether Lck phosphorylation is reduced or not, the change in mean pointing in one direction, the analysis of statistical significance (weakly) in the other.

We thank the reviewer for pointing out that we had overstated the results of the flow cytometry analysis in Fig 3D. We have clarified in the results section that there was a trend towards a decrease in pLck which we may be underpowered to detect. Then in the discussion we have clarified that there was a lack of a significant reduction in pLck changes contrasting to the reduction in TCR signalling and softened the language about the decoupling between IS localisation and pLck Y394 levels. These are mainly highlighted below at line 378-380:

There was, however, a trend towards a decrease in pLck Y394 levels suggesting that there may be some reduction which we were underpowered to detect.

And in the discussion lines 463-466 we softened our conclusions:

Since UNC119 inhibition did not significantly reduce the amount of Y394 activating phosphorylation of LCK we suggest there exists a decoupling between the IS specific localisation of LCK and its activating phosphorylation and demonstrated the requirement in CD8+ T cells for IS localisation of LCK to activate ZAP70

2. The authors should quantify band intensities relative to control in Fig. 4A and B, in particular as gel loading seems uneven. Residual expression of UNC119 in Fig. 5A should also be quantified. The Western blot suggests that the authors have achieved a knock down, not a knockout. This doesn't affect the validity of their conclusions but should nevertheless be acknowledged in the description of the data.

Thank you for pointing out both of these points. The westerns have been quantified and this has been made into supplemental figure 5. We have also modified our language to be more precise about knock down rather than knockout re. the changes in figure 5A. We have made sure to refer to knock-down and depletion rather than knock-out or deletion.

3. It would be good to show all single cell data across the three experimental replicates in Fig. 2K and L (rather than representative data). This can effectively be done using 'SuperPlots' (<https://doi.org/10.1083/jcb.202001064>).

We have adjusted the figures in 2K and N as suggested (we believe that N was meant rather than L for the superplots).

4. The figure caption for Fig. 4 list a panel I (that should be useful) - such panel is not part of the figure though.

Thank you for highlighting this oversight. We have removed the reference to Figure 4I.

Reviewer #2

1. The term "transgenic models" in the title may be misleading, as it typically implies genetically-engineered disease models, which is not the case here. Please consider using a different term for clarity.

Thank you for the comment, we had meant transgenic TCR model to refer to the transgenic mouse line which generates the OTI T cells and had not appreciated the confusion this could cause. We have changed this to refer to primary murine T cells. We hope that this assists in improving clarity. The title now reads as:

2. Please review line 136, as there is an inserted commentary.

Thank you for noticing this. We have removed the commentary.

UNC119 regulates cytotoxic and malignant T cell responses in primary murine T cells and human T acute lymphocytic leukaemia

3. In line 138, please clarify the distinction between UNC119 and UNC119A.

Apologies for the inconsistent naming. Unc119 in humans is also sometimes called Unc119a to differentiate it from Unc119b, however is correctly named Unc119 (Iwasa, Hiroaki, et al. "UNC 119 is a binding partner of tumor suppressor Ras-association domain family 6 and induces apoptosis and cell cycle arrest by MDM 2 and p53." *Cancer Science* 109.9 (2018): 2767-2780.) We have corrected this.

4. Regarding lines 322-323, you state that "UNC119 has a functional role in all subsets of T cells and across different stages of differentiation." Did the single-cell dataset include only mature T cells, or did it also cover precursor/immature cells such as thymic seeding precursor cells and DN cells, representing the full T cell differentiation trajectory? Please clarify this throughout the manuscript.

Thank you for pointing out this lack of precision. We have clarified that these represent mature T cells at the point indicated and also earlier in the paragraph when the dataset is introduced. We have also added in the discussion that we have shown the role of Unc119 in mature T cells rather than in T cells. This occurs multiple times which are highlighted but an example is line 321:

The work presented here highlights that organisation at the intracellular face of the IS is critical for full TCR signal transduction in mature T cells

5. Harmonize the y-axis scales in Figures 1B/C and 1E/F, as inconsistent scales may mislead readers.

We have harmonized the scales across the requested panels.

6. In Figure 4, specify the LCK inhibitor used; if it was dasatinib, the concentration appears high, given its reported efficacy at nanomolar levels (10 to 100 nM) in the literature. Please discuss this and note that the UNC119 inhibitor seems markedly more effective than the LCK inhibitor in reducing cell proliferation, which is perhaps worth highlighting.

The inhibitor used was LCK inhibitor from MedChemExpress (HY-12072) which is (6-(2,6-dimethylphenyl)-2-((4-(4-methylpiperazin-1-yl)phenyl)amino)benzo[4,5]imidazo[1,2-a]pyrimido[5,4-e]pyrimidin-5(6H)-one) rather than dasatinib. The concentration used is that commonly used in the literature. The specific inhibitor has been expanded in the methods and the company and catalogue number added to appropriate figure legends. We have also added a discussion of the relative efficacy of Lck and Unc119 inhibition. Thank you for noticing this oversight in the original manuscript.

This is supported by the fact that inhibition of UNC119 had a greater effect on the *in vitro* proliferation of some patient derived cells. This may be due to the relative efficacy of the inhibitors used in this study or may indicate that inhibition of UNC119 has other, secondary, effects. However, it suggests that UNC119 inhibition may expand the scope of patients who could benefit from therapies targeting LCK.

7. Have you examined the intracellular distribution of UNC119 following inhibitor treatment? Considering the pronounced proliferation inhibition observed, it would be insightful to explore potential mechanisms, particularly as T-ALL cells without TCR engagement do not form an immunological synapse.

We thank the reviewer for sharing their thoughts on this area. We had not investigated the distribution of Unc119 in T-ALL or primary T cells upon inhibition. We have now stained T-ALL cells with or without inhibition to determine whether this gave any insight. We found that inhibition did not alter Unc119 distribution, however, this may not be too surprising as the inhibitor binds the myristoyl binding pocket specifically rather than binding domains driving Unc119 localisation. In figure R1 we present the data from the CCRF cells which show that in DMSO (A) and Unc119i (B) treated CCRF cells there is some Unc119 present in the nucleus and then some perinuclear puncta which matches previously published descriptions of Unc119 distribution. The scale bar is 2um in these figures.

[Figure removed by editorial staff per authors' request].

December 19, 2024

RE: Life Science Alliance Manuscript #LSA-2024-03066-TR

Dr. Ed W Roberts
Cancer Research UK Scotland Institute
Switchback Road
Glasgow, Scotland G61 1BD
United Kingdom

Dear Dr. Roberts,

Thank you for submitting your revised manuscript entitled "UNC119 regulates T cell receptor signalling in primary T cells and T acute lymphocytic leukaemia". We would be happy to publish your paper in Life Science Alliance pending final revisions necessary to meet our formatting guidelines.

- please be sure that the authorship listing and order is correct
- please upload your Table in editable .doc or excel format
- please add ORCID ID for the 3rd corresponding -- they should have received instructions on how to do so
- please add a Summary Blurb/Alternate Abstract and keywords to our system
- titles in the system and the manuscript file must match
- please consult our manuscript preparation guidelines <https://www.life-science-alliance.org/manuscript-prep> and make sure your manuscript sections are in the correct order
- please be sure that authors in the Authors' Contribution section have been correctly entered
- the contributions selected for Payam Gammage do not qualify them for authorship. Please either update the contributions in our system and the Author Contributions section of the manuscript or let us know if the author needs to be removed (and added instead to the acknowledgments section)
- please add your main, supplementary figure, and table legends to the main manuscript text after the references section
- in the Materials & Methods section when describing the mice, please indicate that approval was granted for their use, as well as who granted that approval

A. FINAL FILES:

B. MANUSCRIPT ORGANIZATION AND FORMATTING:

Sincerely,

December 26, 2024

RE: Life Science Alliance Manuscript #LSA-2024-03066-TRR

Dr. Edward W Roberts
Cancer Research UK Scotland Institute
Switchback Road
Glasgow, Scotland G61 1BD
United Kingdom

Dear Dr. Roberts,

Thank you for submitting your Research Article entitled "UNC119 regulates T cell receptor signalling in primary T cells and T acute lymphocytic leukaemia". It is a pleasure to let you know that your manuscript is now accepted for publication in Life Science Alliance. Congratulations on this interesting work.

DISTRIBUTION OF MATERIALS:

Again, congratulations on a very nice paper. I hope you found the review process to be constructive and are pleased with how the manuscript was handled editorially. We look forward to future exciting submissions from your lab.

Sincerely,
